# Intermittent pneumatic compression therapy as a preventive measure for venous thromboembolism after total hip arthroplasty: A systematic review

Vishal Singh[1]☯, Arushi Mishra[1]☯, Deeksha Sharma[1], Umang Singal[2], Najeeb Islam[3], Mohammad Jahirul Islam[ORCID][4], Md. Amran Hossain[5], Md Saifur Rahman[6], Sohel Ahmed[ORCID][7,8]*

**1** Department of Physiotherapy, Maharishi Markandeshwar University, Kumarhatti, Solan, Himachal Pradesh, India, **2** Department of General Surgery, United Institute of Medical Sciences, Prayagraj- UP, India, **3** Department of Physiotherapy, Revive Physiotherapy Clinic, Bhopal, India, **4** Department of Physical Medicine and Rehabilitation, MAG Osmani Medical College Hospital, Sylhet, Bangladesh, **5** Department of Physiotherapy, Women's and Children's General Hospital, Dhaka, Bangladesh, **6** Department of Physiotherapy and Rehabilitation, Jashore University of Science and Technology, Jashore, Bangladesh, **7** Ahmed Physiotherapy & Research Center, Kalabagan, Dhaka, Bangladesh, **8** Directorate of Student's Welfare, Bangladesh University of Engineering and Technology, Dhaka, Bangladesh

☯ These authors contributed equally to this work and share first authorship.
* ptsohel@gmail.com

## Abstract

### Background

Venous thromboembolism (VTE) is a significant and avoidable complication that may occur after total hip arthroplasty (THA). Various mechanical and chemical prophylactic measures may mitigate this elevated risk of death and functional impairment. Consequently, early prevention of VTE is essential via the identification of related risk factors.

### Methods

A search was performed using the databases of PubMed, ScienceDirect, PEDro, and Cochrane Library to get papers from 2004 to 2024 in accordance with PRISMA guidelines. Only randomized controlled trials (RCTs) published in English that included at least one group undergoing intermittent pneumatic compression (IPC) treatment as a prophylactic intervention after total hip arthroplasty (THA) were included. This systematic review has been registered in PROSPERO. The quality evaluation of the included studies was conducted using the PEDro scale and the Cochrane risk of bias instrument.

### Result

We selected 12 studies from a total of 733 based on predetermined criteria. A total of 2,352 patients of both genders underwent total hip arthroplasty, comprising 1,294

**Data availability statement:** All relevant data are within the manuscript and its supporting information files.

**Funding:** The author(s) received no specific funding for this work.

**Competing interests:** The authors have declared that no competing interests exist.

**Abbreviations:** THA, total hip arthroplasty; VTE, venous thromboembolism; DVT, deep vein thrombosis; PE, pulmonary embolism; IPC, intermittent pneumatic compression; PICO, participant, intervention, comparison, outcome; CT, computer tomography; NPRS, numeric pain rating scale; VAS, visual analogue scale; GCS, graded compression stocking; RCT, randomized controlled trail; AAOS, American Academy of Orthopaedic Surgeons; ACCP, American College of Chest Physicians.

patients in the experimental group and 1,058 patients in the control group across the included studies. The results indicate that the combination of IPC and pharmaceutical agents was the most effective treatment for reducing VTE risk in patients who underwent THA.

## Conclusion

IPC therapy is very effective in avoiding VTE, particularly when used in combination with pharmacological therapies after THA surgery. The best ways to lower the risk of VTE are to use both IPC and anticoagulants together. However, IPC alone may lower the risk of VTE compared to not using any prevention at all. In general, IPC is a crucial component of comprehensive VTE prevention strategies in THA.

## Introduction

Hip arthroplasty is the most common surgical intervention for addressing hip joint disorders, which includes hemiarthroplasty and total hip arthroplasty (THA). Total hip arthroplasty yields superior results for patient functionality and quality of life, as well as improved clinical outcomes [1]. THA involves replacing both articular surfaces of the hip with prosthetic implants, which consist of the acetabular component, the femoral component, and the bearing surfaces fitted into their natural positions with or without cement [2]. Annually, more than 1 million hip replacement procedures are conducted worldwide [3]. Despite the global success of THA surgery, it is associated with systemic and procedure-specific complications like venous thromboembolism (VTE), hematoma formation, nerve injury, fracture, postoperative dislocation, deep infection, wound, heterotopic ossification, and other arthroplasty-related issues [4,5]. Among these problems, VTE is a prominent and serious concern in individuals who have undergone THA. It includes deep vein thrombosis (DVT) and pulmonary embolism (PE), primarily resulting from endothelial damage, a hypercoagulable condition, and venous stasis [6,7]. Additional potential risk factors for the development of in-hospital VTE encompass advanced age (>70 years), female sex, obesity, diabetes mellitus, cardiovascular illness, revision surgery, cement fixation, hemorrhage, and a prior history of VTE [8].

The prevalence of VTE following THA ranges from 40% to 80%, with a median occurrence of 0.6% (DVT approximately 0.24% and PE around 0.41%), potentially up to 2.5% in revision THA [9]. Thus, the timely prevention of VTE with recommended prophylaxis is essential to prevent additional complications, such as post-thrombotic syndrome and pulmonary embolism [7,10,11]. Various approaches exist for the treatment of VTE, with the most prevalent being chemical prophylaxis, which encompasses the use of various anticoagulants, and mechanical prophylaxis, which involves the application of devices such as graduated compression stockings, elastic stockings, and foot pumps [12,13]. Chemical prophylaxis is linked to bleeding and wound complications, while mechanical prophylaxis is devoid of such side effects [13]. Conversely, numerous research studies and guidelines indicate that the

synergistic effect of both mechanical and chemical prophylaxis is more advantageous than the use of either method alone [14,15].

The intermittent pneumatic compression (IPC) device features an inflatable sleeve encircling the calf or foot, which is linked to an electrical pneumatic pump that inflates the sleeve with air. This exerts external pressure on the leg or foot, compressing the deep vein, enhancing blood circulation, and reducing the formation of blood clots, hence diminishing the risk of VTE [13,16]. Researchers conducted numerous clinical studies to evaluate the effectiveness of IPC devices as a safe and preferable approach for mitigating VTE following major orthopedic surgeries. Despite being a non-invasive therapy, researchers have not thoroughly studied the effectiveness of IPC as a stand-alone or supplemental intervention for the prevention of VTE in patients with THA. The aim of this systematic study was to assess the efficacy of IPC in mitigating the risk of VTE in patients undergoing THA. This evaluation may assist clinicians in enhancing postoperative care and could offer a safer, non-pharmacological alternative to anticoagulants for IPC therapy.

## Methods

This research adheres to the Preferred Reporting Items for Systematic Review and Meta-Analysis (PRISMA) criteria and presents the required information appropriately [17]. This systematic review has been registered in PROSPERO under registration number CRD42024553081.

### Search strategy

Electronic databases including PubMed, Web of Science, Science Direct, Cochrane Library, and PEDro, were collectively searched to assess evidence regarding IPC for the prevention of VTE in patients with post-THA, restricting the search to studies published in English from 2004 to 31 August 2024. Boolean terms were used with keywords like "total hip arthroplasty," "intermittent pneumatic compression," "mobile compression device," "post-surgical complications," "venous thromboembolism," "deep vein thrombosis," and "pulmonary embolism," as seen in Table 1.

### Eligibility criteria

The studies were chosen according to predetermined selection criteria based on PICO (participants, intervention, comparison, outcomes) as shown in Table 2. This analysis encompassed papers from both male and female patients, aged 40–80 years, who underwent THA. The treatments included IPC devices either alone or in conjunction with anticoagulants, whereas the outcome measurements encompassed Duplex or Doppler ultrasonography, CT angiography, girth measurement, NPRS scale/VAS, and bleeding evaluation. This review excluded studies including preoperative patients receiving just pharmaceutical therapy as an intervention, as well as randomized controlled trial designs published in languages other than English.

### Selection process

Two independent researchers performed a literature review employing pre-established research criteria. Subsequent to the removal of duplicate articles from the chosen studies, the researchers implemented a screening procedure by assessing the abstract and title of each study article prior to a comprehensive examination of the full text. Twelve articles that satisfied the review's inclusion criteria were subsequently incorporated.

### Data extraction process

After analyzing the selected articles, data extraction process was completed using a Microsoft Excel spreadsheet. The accompanying data were extracted from each included research: 'first author name,' 'age of study participants,' 'participants in experimental and control groups,' 'intervention,' 'outcome measures and study conclusion.' No author was consulted throughout the data extraction process.

**Table 1. Search strategy with keywords.**

| Keywords | Boolean Terms | PubMed | ScienceDirect | Cochrane Library | PEDro | Total |
|---|---|---|---|---|---|---|
| #1 Total Hip Arthroplasty<br>#3 Intermittent Pneumatic Compression | #1 AND #3 | 2 | 50 | 21 | 1 | 74 |
| #1 Total Hip Arthroplasty<br>#4 Mobile Compression Device | #1 AND #4 | 2 | 24 | 7 | 0 | 33 |
| #1 Total Hip Arthroplasty<br>#2 Complications<br>#5 Venous thromboembolism | #1 AND #2 AND #5 | 25 | 261 | 132 | 1 | 422 |
| #1 Total Hip Arthroplasty<br>#3 Intermittent Pneumatic Compression<br>#5 Venous thromboembolism | #1 AND #3 AND #5 | 4 | 48 | 8 | 0 | 60 |
| #6 Deep Vein Thrombosis<br>#3 Intermittent Pneumatic Compression<br>#1 Total Hip Arthroplasty | #1 AND #3 AND #6 | 2 | 39 | 11 | 10 | 62 |
| #1 Total Hip Arthroplasty<br>#3 Intermittent Pneumatic Compression<br>#7 Pulmonary embolism | #1 AND #3 AND #7 | 1 | 37 | 8 | 4 | 50 |
| #1 Total Hip Arthroplasty<br>#7 Pulmonary embolism | #1 AND #7 | 20 | 0 | 0 | 0 | 0 |
| #7 Pulmonary embolism<br>#3 Intermittent Pneumatic Compression | #7 AND #3 | 15 | 0 | 0 | 0 | 0 |

**Table 2. Eligibility criteria based on PICO format.**

| Criteria | Inclusion | Exclusion |
|---|---|---|
| Population | Both genders undergoing total hip arthroplasty | Pre-operative patients with any hip pathology |
| Interventions | IPC device alone or with any other treatment | Only pharmacological treatment/ Conservative Management |
| Comparison | Other mechanical prophylaxis | Any other treatment |
| Outcome measures | Duplex or Doppler ultrasound, pulmonary CT angiography, Girth measurement, NPRS scale, pain VAS and bleeding assessment. | CBC, MRI, |

## Methodological quality

Two independent evaluators appraised the study quality using the 'Pedro scale' and the 'Cochrane risk of bias instrument'. The PEDro assessment scale has 11 points, with a maximum attainable score of 10 points. The final score calculation excludes the first variable, which denotes the qualifying conditions. Every answer receives a score of either 1 or 0. A score ranging from 0 to 3 indicates "poor quality," 4–5 signifies "fair," 6–8 represents "good," and > 9 defines "excellent" [18]. The "Cochrane risk of bias tool" assesses study quality by inquiring about random allocation, used treatments, unexamined outcomes, randomly assigned outcomes, and other biases present in published research. Three categories for the study were established based on these inquiries: low, medium, and high-risk bias [19].

## Results

### Study selection

A PRISMA flowchart illustrates the process of research selection. Utilizing several data search engines, we identified a total of 733 articles. Upon eliminating the duplicates, 431 articles remained. Seventeen articles were eliminated based

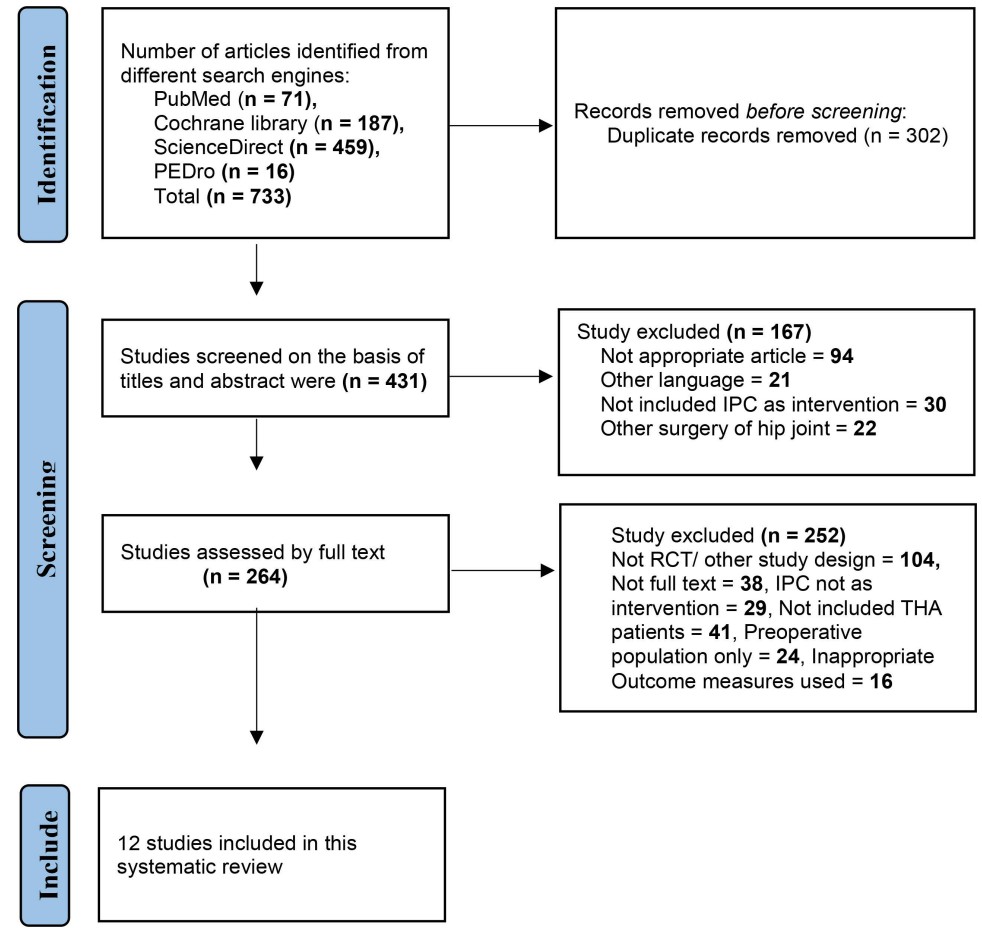

**Fig 1. PRISMA flow chart of the study.**

on the title and abstract. A total of 264 (S1 Text) papers were evaluated based on full text/open access, and ultimately, 12 studies [6,7,10,20–28] were included into this systematic review, as seen in Fig 1. The PRISMA checklist is used to ensure transparent and comprehensive reporting of systematic reviews (S2 Text).

## Methodological quality

**PEDro scale.** The PEDro scale was used to evaluate the quality of the included studies. Two investigators computed the PEDro score for all included studies manually. Ten papers [6,10,20,21,23–28] in this review are classified as 'good' quality, whereas two articles [7,22] are deemed of fair quality, as shown in Table 3.

## Cochrane risk of bias

The Cochrane risk of bias tool was used to assess the quality of the research, as seen in Fig 2. The publications were classified into three risk categories: low, medium, and high. Of the 12 trials, 10 [6,10,20,21,23–28] demonstrate a minimal risk of bias regarding allocation concealment. Regarding participant blinding and outcome assessment blinding, almost every research exhibits a significant risk of bias. Regarding random sequence generation, eight studies [6,10,20,23,24,26–28] were included, whereas concerning incomplete outcome data, all 12 studies [6,7,10,20–28] shown

Table 3. Studies quality assessment based on the PEDro score.

| Author/year | 1 | 2 | 3 | 4 | 5 | 6 | 7 | 8 | 9 | 10 | 11 | Total |
|---|---|---|---|---|---|---|---|---|---|---|---|---|
| Kwak. S et al 2016 [7] | 1 | 0 | 0 | 1 | 0 | 0 | 0 | 1 | 1 | 1 | 1 | 5 |
| Paudel. S et al 2019 [10] | 1 | 1 | 1 | 1 | 0 | 0 | 0 | 1 | 1 | 1 | 1 | 7 |
| Pellino. C et al 2023 [6] | 1 | 1 | 1 | 1 | 0 | 0 | 0 | 1 | 1 | 1 | 1 | 7 |
| Wang. D et al 2018 [22] | 1 | 0 | 0 | 1 | 0 | 0 | 0 | 1 | 1 | 1 | 1 | 5 |
| Colwell. W et al 2010 [23] | 1 | 1 | 1 | 1 | 0 | 0 | 0 | 1 | 1 | 1 | 1 | 7 |
| Liqun. W et al 2020 [24] | 1 | 1 | 1 | 1 | 0 | 0 | 0 | 1 | 1 | 1 | 1 | 7 |
| Silbersack.Y. et al 2004 [25] | 1 | 1 | 0 | 1 | 0 | 0 | 0 | 1 | 1 | 1 | 1 | 6 |
| Eiselc. R et al 2007 [26] | 1 | 1 | 1 | 1 | 0 | 0 | 0 | 1 | 1 | 1 | 1 | 7 |
| Khalafallah. A et al 2018 [27] | 1 | 1 | 1 | 1 | 0 | 0 | 1 | 1 | 1 | 1 | 1 | 8 |
| Paudel. S 2014 [28] | 1 | 1 | 1 | 1 | 0 | 0 | 0 | 1 | 1 | 1 | 1 | 7 |
| Ben-Galim P. et al 2009 [20] | 1 | 1 | 1 | 1 | 0 | 0 | 0 | 1 | 1 | 1 | 1 | 7 |
| Yokote R. et al 2011 [21] | 1 | 1 | 0 | 1 | 0 | 0 | 0 | 1 | 1 | 1 | 1 | 6 |

List of items: 1 – Eligibility criteria, 2 – Randomly allocated, 3 – Concealed allocation, 4 – Baseline similarity, 5 – Blinding of subject, 6 – Blinding of therapist, 7 – Blinding of assessors, 8 – Measures of at-least one key outcome, 9 – Control group received treatment, 10 – Inter-group statistical, 11 – The study provides both point variability.

a minimal risk of bias. In all papers examined, the author did not acknowledge any kind of reporting bias or other biases. Consequently, there was an ambiguous risk bias concerning two variables: inadequate data outcomes and reporting bias.

## Study and population characteristics

The data presented in Table 4 illustrates a notable consistency in the study and participant characteristics among the articles reviewed. All studies included were randomized controlled trials conducted in Korea [7] with additional trials carried out in India [10], Switzerland [6], China [22,29], Germany [25,26], the USA [23], Australia [27], Nepal [28], Japan [21], and Israel [20]. This analysis included 2,356 patients undergoing total hip arthroplasty, with 1,298 assigned to the experimental group and 1,058 to the control group. The prevalent outcome measures identified in the 12 articles analysed include duplex ultrasound, colour doppler ultrasound, pulmonary CT angiography, girth measurement, NPRS scale, pain visual analog scale, and bleeding assessment [6,7,10,20–28].

## Study intervention

Each study featured in this systematic review [6,7,10,20–28] incorporates at least one group where IPC is utilized as an intervention. The intervention group included 206 [26] participants in the IPC alone category, 617 [6,7,10,23,28] in the IPC with aspirin category, 348 [6,21,22] in the IPC with graduated stockings category, 342 [20,24–26] in the IPC with LMWH category, 168 in the IPC with enoxaparin category, and 167 in the IPC with rivaroxaban category [27].

While all intervention groups incorporated IPC, there was significant variability in the usage time, frequency, pressure, inflation-deflation duration, and overall treatment length. The treatment duration across all studies ranges from postoperative day 1 to postoperative day 16. Four studies suggest a usage duration for IPC between 6 and 8 hours daily [22–24,26]. Conversely, Eisele R et al [26] indicates a usage duration ranging from 2 to 10 hours daily. Paudel S et.al and Paudel S [10,28] support the implementation of IPC continuously throughout the day until mobilization occurs. Kwak HS et.al and Silbersack Y et.al [7,25] in their trial, indicate a continuous application of IPC whereas Wang D et.al, Liqun W et.al and Khalafallah A et.al [22,24,27] suggest continuous application ranges between 18–48 hours, and Carnevale PV et.al [6] recommends a 30-minute application period. The inflation pressure settings for the cuff vary between 45 and 100 mmHg.

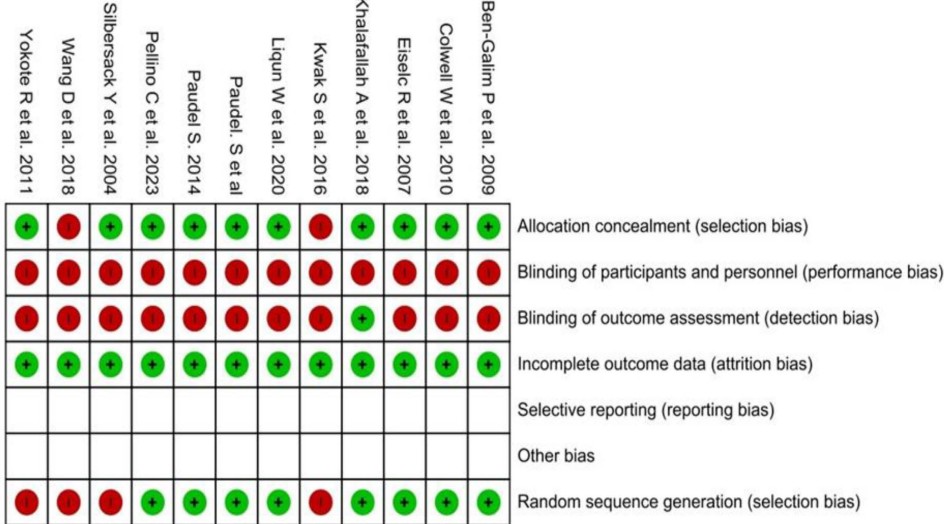

**Fig 2. Cochrane risk of bias of all included studies.**

Two studies recommended a pressure setting of 45 mmHg [22,26], while two others suggested a pressure of 52 mmHg [25,26] for distal cuff inflation. One study established a pressure of 50 mmHg [23], and another aimed for a pressure setting of 100 mmHg [6]. Six studies [6,7,22,23,25,26] implemented a treatment duration of 10 days, while two studies (8,24) utilized IPC until the 4th post-operative day. Additionally, two studies [23,26] suggested a treatment duration exceeding 10 days, one study [20] applied IPC for 5 days, and another study [21] applied IPC for 2 days.

## Discussion

This systematic review aimed to examine the efficacy and reliability of IPC devices used as a safer treatment option for VTE risk reduction after THA surgery. Twelve randomized controlled trials were considered in this review and examined according to predetermined inclusion criteria to derive a clear result. The PEDro scale and the Cochrane risk of bias tool facilitated the quality evaluation of the included studies. This review had 2,356 participants of both sexes, aged 40–80 years. A total of 1,298 patients comprised the experimental group, whereas 1,058 patients constituted the control group. All patients underwent IPC as an intervention, either singularly or in conjunction with pharmacological treatments, to avert VTE following total hip arthroplasty. Moreover, the majority of research contrasted IPC with anticoagulation concerning significant clinical outcomes and encapsulated the principal findings derived from this comparison [6,7,10,20,21,23–28].

The primary outcome of this review was VTE, which was analysed individually in each article to assess the efficacy of IPC according to the available data. Among the 12 articles reviewed, Woo-Lam Jo et al. [11] endorsed the application of duplex ultrasound as an outcome measure for DVT detection which is also supported in other 4 articles [7,21,23,26]. In contrast, H. Al-Thani et.al and Wong Lin et.al [30,31] presented evidence supporting the use of colour doppler ultrasonography supported by other 7 articles [10,20,22,24,25,27,28]. Both AAOS (American Academy of Orthopedic Surgeons) and ACCP (American College of Chest Physicians) recommend doppler or duplex ultrasonography for the screening of VTE post-operatively and during the time of discharge [32,33]. The studies conducted by Hogg K et al. [34] and Albrecht MH et al. [35] provided justification for the widespread application of CT angiography in the detection of PE which is also in line with other five studies [7,10,25,27,28]. Six investigations [6,21,22,24,27,28] evaluated the risk of bleeding taken into account as a baseline risk of VTE reoccurrence, which was further elaborated in an another study by Nopp. S et al [36].

**Table 4. Study and population characteristic.**

**Characteristic of 12 included studies**

| S.No. | Author & Year | Age | Participants | Interventions | Outcome Measures | Conclusion/ Main finding |
|---|---|---|---|---|---|---|
| 1. | Kwak. S et al 2016 [7] | EG = 54 years<br>CG = 53 years | EG = 233 patients (106 M and 127 F)<br>CG = 146 patients (80 M and 66 F) | EG – IPC plus low dose aspirin was used.<br>Use time = NS<br>Pressure = NS<br>Duration = 10 days<br>CG – only low dose aspirin was used. | Homans' sign and CT angiogram Duplex ultrasonogram | The incidence of VTE was much lower in the EG group (1.3%) than in CG group (4.1%).<br>This study suggest that IPC might be effective and safe for the prevention of postoperative VTE. |
| | Paudel. S et al 2019 [10] | EG = 49.08 years<br>CG = 48.26 years | EG = 90 patients (51 M and 39 F)<br>CG = 90 patients (48 M and 42 F) | EG – Aspirin plus IPC<br>Use time = NS<br>Pressure = NS<br>Duration = till 4th POD<br>CG- low molecular weight heparin (LMWH) | Color doppler ultrasonography and pulmonary CT angiography | The rate of major bleeding events was 1.11% in EG group and 10% in CG group. |
| 3 | Pellino. C et al., 2023 [6] | EG = 68.38 ± 9.35 years<br>CG = 68.09 ± 9.11 years | EG = 24 patients (9 M and 15 F)<br>CG = 23 patients (8 M and 15 F) | EG – IPC combined with standard VTE therapy<br>Use time = 30 minutes<br>Pressure = 100 mmHg<br>Duration = 10 days<br>CG – standard VTE therapy (medications, compression stocking,) | Limb girth measurement, Numeric pain rating scale (NPRS), and 20-meter walk test | The combination of standard VTE therapy with IPC is more effective than the standard treatment alone. |
| 4 | Wang. D et al., 2018 [22] | EG = 55.6 ± 14.2 years<br>CG = 50.6 ± 19.2 years | EG = 51 patients (29 M and 22 F)<br>CG = 61 patients (36 M and 25 F) | EG – IPC device and stocking<br>Use time = 24 hrs on POD 1 then 6–8 hrs a day<br>Pressure = 45 mmHg<br>Duration = 10 days<br>CG = GCS | Doppler sonography | There was significant reduction of VTE in EG group (3.92%, 2/51) compared to CG group (9.84%, 6/61). |
| 5 | Colwell jr. W et al., 2010 [23] | EG = 63 years<br>CG = 62 years | EG = 198 patients (55% F)<br>CG = 194 patients (54% F) | EG - mobile compression device (MCD) including IPC and venous foot pump combined with aspirin<br>Use time = 6 hrs a day<br>Pressure = 50 mmHg<br>Duration = 10–12 days<br>CG – LMWH only | Duplex ultrasonography | Major bleeding events occurred in 11 patients (6%) all in the CG group.<br>VTE reported in 5.1% in EC group and 5.3% in CG group. |
| 6 | Liqun. W et al., 2020 [29] | EG = 60.6 years<br>CG = 63.1 years | EG = 47 patient (18 M and 29 F)<br>CG = 47 patients (13 M and 34 F) | Both groups receive IPC and LMWH after returning to the ward<br>EG – ICP used for 6 hours<br>Pressure = 10 Kpa<br>Duration = POD 1<br>CG – IPC used for 18 hours | Pain visual ana-log scale Pittsburgh Sleep Assessment Scale Doppler Ultrasound Girth measurement Bleeding assessment | Short-term application of IPC after THA will not increase degree of pain, calf swelling, and DVT. It can also improve the quality of sleep and facilitate the recovery of the limb. |
| 7 | Silber-sack.Y et al 2004 [25] | EG = 63 years<br>CG = 65 years | EG = 33 patients<br>CG = 28 patients | EG = IPC combined with LMWH<br>Use time = NS<br>Pressure = 52 mmHg<br>Duration = 10 days<br>CG = LMWH plus GCS | Ultrasonography | LMWH with IPC show more effectiveness than LMWH used with GCS. |
| 8 | Eisele R. et al 2007 [26] | EG = NS<br>CG = NS | EC = 190 patients<br>CC = 116 patients | EG = IPC with LMWH<br>Use time = 2–10 hrs a day<br>Pressure = 45–52 mmHg<br>Duration = 1–16 days<br>CG = LMWH alone | Duplex color-coded ultrasound | No any case of DVT in EC group while in CG 5.2% (6 patients) develop DVT. |

*(Continued)*

**Table 4.** (Continued)

| | | | | Characteristic of 12 included studies | | |
|---|---|---|---|---|---|---|
| 9 | Khalafallah. A et al. 2018 [27] | EG = 67.3 years CG = 67.2 years | EG = 168 patients CG = 167 patients | EG = IPC plus Enoxaparin CG = IPC plus Rivaroxaban Use time = 24 hrs and 48 hrs Pressure = NS Duration = 1 and 2 days | Doppler Ultrasound CT angiography | After 6-month POD follow up period, application of IPC followed by chemical prophylaxis was effective in VET management with no increased bleeding risk. |
| 10 | Paudel. S et al 2014[28] | EG = 48 years CG = 49 years | EG = 72 patients CG = 78 patients | EG = IPC plus aspirin CG = LMWH Use time = NS Pressure = NS Duration = till POD 4 | Duplex ultrasound CT angiography Chest X-ray | the risk of major and minor bleeding events with the use of asprin plus IPC is lower in compression of low molecular weight heparin |
| 11 | Ben-Galim P. et al 2009 [20] | EG => 60 years CG => 60 years | EG = 25 patients CG = 25 patients | EG = WizAir-DVT IPC device Use time = NS Pressure = 50 mmHg Duration = 5 days CG = IPC plus LMWH Use time = NS Pressure = 35–55 mmHg Duration = 5 days | Doppler Ultrasound | The WizAir-DVT IPC device shows more effective and safer than routinely used Kendall SCD device. It also provides easier postoperative patients mobilization and early hospital discharge. |
| 12 | Yokote R. et al 2011[21] | EG-1 = 64 years EG-2 = 63 years CG = 63 years3 | EG-1 = 83 patients EG-2 = 84 patients CG = 83 years | Patients in all group receive IPC and GCS Use time = NS Pressure = NS Duration = 2 days EG-1 = Enoxaparin EG-2 = Fondaparinux CG = Placebo | Duplex ultrasound Bleeding assessment | The rate of VET was 7.2% with placebo, 7.1% with fondaparinux, and 6% with enoxaparin. This study suggest that application of IPC shows more effectiveness and safety regardless of anticoagulant drugs alone. |

**Abbreviations used:** EG – experimental group, CG- control group, NS – not specified, M – male, F – female, VTE – venous thromboembolism, IPC – intermittent pneumatic compression therapy, LMWH – low molecular weight heparin, GCS – graduated compression stocking, SCD – sequential compression device, POD – post-operative day

As a mechanical preventative measure, IPC helps lower venous congestion and stasis by squeezing the lower limb. It also helps lower pro-coagulant and thrombolysis. The guidelines of AAOS and ACCP, which support the use of either mechanical or chemical prophylaxis, or both, recommend IPC as an effective modality to significantly reduce the risk of VTE [32,33,37]. Nevertheless, the characteristics of IPC devices exhibit significant diversity across various studies and may demonstrate divergent effects depending on use duration, frequency, and compression pressure. Three research [22,23,26] recommend 6–8 hours of intermittent pneumatic compression (IPC) treatment over a duration of 10 days, while two additional studies [7,25] advocate for continuous IPC administration for the whole 10-day period. Two studies [10,28] use intermittent pneumatic compression (IPC) continuously throughout the day until the patient mobilizes on the fourth postoperative day. Carnevale PV et.al [6] in their study administered IPC for 30 minutes, twice day, for a duration of 10 days. Khalafallah A et.al [27] indicated that a 24-hour use of IPC, with pharmaceutical prophylaxis, successfully reduced VTE risk without elevating bleeding risk. Liqun W et. al [24] compared 6 hours of IPC use with 18 hours, indicating that 6 hours is more beneficial. Ben-galim P et. al [20] employs WizAir-DVT IPC devices alongside standard IPC devices for a duration of 5 days, concluding that WizAir-DVT IPC devices provide enhanced patient movement and prompt hospital release. Our results corroborate the ACCP's guideline that all THA patients should get thromboprophylaxis using IPC, pharmacological medications, or a combination of both, for a duration of 10–14 days, extending up to 35 days [32].

Among 12 investigations, five studies [6,7,10,23,28] use IPC in conjunction with aspirin, four studies [20,24–26] utilize IPC with LMWH, and two studies [21,27] include alternative anticoagulants such as enoxaparin and rivaroxaban alongside IPC. In three included RCT trials [6,21,22], the author sought to evaluate the outcomes of graded compression stockings (GCS) as standard treatment with intermittent pneumatic compression (IPC) and found that IPC combined with GCS significantly reduces venous thromboembolism (VTE) compared to traditional GCS therapy alone.

This review included 12 studies that recommended IPC as a thromboprophylaxis to reduce bleeding and VTE risk. Previous studies by Zareba P. et al. [14] and Kakkos S. et al. [38] supported the clinical findings of these studies, suggesting that IPC, compared with other interventions, has decreased the risk of VTE and bleeding complications, and that a combination of IPC and medications was more effective than medications alone.

### Strength and limitations of the study

This systematic review consolidates data from multiple trials, providing a thorough assessment of the efficacy of IPC while improving the generalizability of the findings by reducing bias through quality evaluation. This study supports the practical application of IPC in post-THA care by demonstrating that it serves as a non-invasive, clinically significant alternative or complement to pharmacological prophylaxis, particularly for patients at elevated risk of VTE or those unable to get anticoagulant therapy. This comprehensive review has various limitations, notably a significant variation in IPC frequency, dosage, and session, which obstructs the determination of a specific IPC protocol for patients who underwent THA. Our review is confined to articles published in English, exclusively comprising RCT trials from 2004 to 2024. Although pulmonary embolism is frequently reported in venous thromboembolism patients and associated with a significant mortality risk, the research included in this systematic review failed to offer adequate information regarding its clinical effects. The absence of comprehensive randomized studies in this domain underscores the necessity for further research.

### Clinical implication

This systematic review consolidates data from multiple studies, providing a thorough analysis of the efficacy of IPC while improving the generalizability of the findings by reducing bias through quality assessment. This study illustrates the efficacy of IPC as a non-invasive, clinically relevant alternative or complement to pharmacological prophylaxis in post-THA management, especially for patients predisposed to VTE or contraindicated for anticoagulant therapy.

### Conclusion

The systematic review highlights the potential advantages of IPC treatment in avoiding VTE after THA. IPC therapy appears to be a useful preventive approach based on the available data, especially when combined with pharmaceutical treatments. The most effective preventive measures to reduce the risk of VTE include combinations of IPC and anticoagulants, even though IPC alone can reduce the incidence of VTE when compared to no prophylaxis. In general, IPC continues to be an important part of comprehensive VTE prevention strategies in THA, with possibilities for improvements in clinical protocols and personalized care methods.

### Supporting information

**S1 Text. List of included and excluded studies.**
(DOCX)

**S2 Text. PRISMA _ Checklist.**
(DOCX)

## Acknowledgments

The article was presented as a paper presentation at the 3rd International Conference on Advances in Physiotherapy and Rehabilitation (AIPR 24) at Chitkara University, Rajpura, Punjab, where it won second place in the junior category. The authors express gratitude to Maharishi Markandeshwar University, Kumarhatti-Solan, Himachal Pradesh, for its logistical support.

## Author contributions

**Conceptualization:** Vishal Singh, Arushi Mishra, Deeksha Sharma, Umang Singal, Najeeb Islam, Mohammad Jahirul Islam, Md. Amran Hossain, Md Saifur Rahman, Sohel Ahmed.

**Data curation:** Vishal Singh, Deeksha Sharma, Md Saifur Rahman.

**Formal analysis:** Vishal Singh, Arushi Mishra.

**Investigation:** Vishal Singh, Arushi Mishra, Deeksha Sharma, Umang Singal, Najeeb Islam, Md Saifur Rahman.

**Methodology:** Vishal Singh, Arushi Mishra, Deeksha Sharma, Umang Singal, Najeeb Islam, Mohammad Jahirul Islam, Md. Amran Hossain, Sohel Ahmed.

**Project administration:** Arushi Mishra, Najeeb Islam, Md Saifur Rahman.

**Resources:** Arushi Mishra, Deeksha Sharma, Umang Singal, Najeeb Islam, Md Saifur Rahman.

**Software:** Arushi Mishra, Najeeb Islam, Mohammad Jahirul Islam.

**Supervision:** Arushi Mishra, Deeksha Sharma, Mohammad Jahirul Islam, Md. Amran Hossain, Sohel Ahmed.

**Validation:** Arushi Mishra, Umang Singal, Najeeb Islam, Mohammad Jahirul Islam, Md Saifur Rahman, Sohel Ahmed.

**Visualization:** Arushi Mishra, Umang Singal, Najeeb Islam, Mohammad Jahirul Islam, Md Saifur Rahman, Sohel Ahmed.

**Writing – original draft:** Vishal Singh, Arushi Mishra, Deeksha Sharma, Umang Singal, Mohammad Jahirul Islam, Md Saifur Rahman, Sohel Ahmed.

**Writing – review & editing:** Najeeb Islam, Md. Amran Hossain, Sohel Ahmed.

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
