## [Decision Letter · Decision Letter 0]

8 Dec 2024

PONE-D-24-45814Intermittent pneumatic compression therapy as a preventive measure for venous thromboembolism after total hip arthroplasty: A systematic reviewPLOS ONE

Dear Dr. Ahmed,

Thank you for submitting your manuscript to PLOS ONE. After careful consideration, we feel that it has merit but does not fully meet PLOS ONE’s publication criteria as it currently stands. Therefore, we invite you to submit a revised version of the manuscript that addresses the points raised during the review process.

We look forward to receiving your revised manuscript.

Kind regards,

Amirhossein Ghaseminejad-Raeini

Academic Editor

PLOS ONE

Journal Requirements:

Reviewers' comments:

Reviewer's Responses to Questions

**Comments to the Author**

1. Is the manuscript technically sound, and do the data support the conclusions?

Reviewer #1: Yes

Reviewer #2: No

2. Has the statistical analysis been performed appropriately and rigorously? 

Reviewer #1: Yes

Reviewer #2: No

3. Have the authors made all data underlying the findings in their manuscript fully available?

Reviewer #1: Yes

Reviewer #2: Yes

4. Is the manuscript presented in an intelligible fashion and written in standard English?

Reviewer #1: Yes

Reviewer #2: No

5. Review Comments to the Author

Reviewer #1: At the end I have some questions about this article.

The specific questions and recommendations are as follows:

1:Conclusion parts in both the abstract and the main text needs to be rewritten and lengthened. Consider

adding more supporting evidence.

2:The discussion should be more comprehensive. Furthermore, The introduction presents an appropriate

outline but it is better to be lengthened. Adding additional references is recommended in both introduction

and discussion section.

3:It is recommended to add a new section at the end of the article after conclusion section, describing any

abbreviation which is being used throughout the article.

Thanks to the authors for their hard work on this Study. It’s a pleasure to review such a contribution to the field.

Reviewer #2: I am glad to have the opportunity to review this manuscript and thank the authors for conducting this research on the use of intermittent pneumatic compression therapy as a preventive measure for venous thromboembolism after total hip arthroplasty. The results section lacks clarity on the added value of the study to the topic. The authors should provide conclusive insights from the included studies and consider presenting new findings. Additionally, the discussion requires substantial revision. My specific comments are as follows:

Line 51: The authors should provide a full explanation of "THA" when it is first mentioned in the manuscript.

Line 53: Consider replacing the word “lakh” with a term more universally understood by the audience.

Line 55: The authors are encouraged to explain "VTE" the first time it appears in the manuscript.

Line 71: Similarly, please provide an explanation for "IPC" at its first mention in the manuscript.

Line 91: Could the authors clarify the rationale behind restricting the search dates to studies published from 2004 onward?

Line 94: The complete search strategy should be included as a supplementary file and referenced in the corresponding paragraph of the manuscript. If this is not feasible, please provide it in your response to comments.

Line 105: The authors state that RCTs were excluded; however, the results section mentions that all included articles were RCTs. This discrepancy suggests an error in line 105 that needs correction.

Line 109: In the “Selection Process” section, please specify how differences between the two reviewers were resolved.

Line 150, Figure 2: The authors should address the selective bias and reporting bias components in Figure 2 for completeness.

Table 4: It would enhance the table's informativeness if study design, country, and follow-up duration were included.

Table 4: Two of the included studies were conducted by Paudel et al. The authors are advised to verify whether these two articles share participants. If they do, only the more comprehensive study should be included unless the outcomes reported are distinct.

Line 183: "LMWH" should be explained the first time it is mentioned in the manuscript.

Results Section: The results section lacks clarity on the added value of the study to the topic. The authors should provide conclusive insights from the included studies and consider presenting new findings, such as the VTE rate following IPC, using proportion analysis or pooled prevalence analysis.

Line 227: The definition of IPC appears in the discussion section but should be introduced earlier, in the introduction, where IPC is first mentioned.

Discussion Section: The discussion requires substantial revision. Much of its content (e.g., Lines 208-214, Lines 216-223, Lines 224-246) would be more appropriate in the results section. The discussion should focus on synthesizing the findings, their implications, and the broader context of the study.

General Comment: The manuscript would benefit from a thorough revision of its English for clarity and readability.

6. PLOS authors have the option to publish the peer review history of their article (what does this mean? ). If published, this will include your full peer review and any attached files.

**Do you want your identity to be public for this peer review?** For information about this choice, including consent withdrawal, please see our Privacy Policy .

Reviewer #1: **Yes: ** SEYED AMIRHOSSEIN MAZHARI

Reviewer #2: No

---

## [Author Response · Author response to Decision Letter 1]

17 Dec 2024

We express our gratitude to the reviewer for their meticulous and in-depth reading of this manuscript, as well as for their insightful remarks and helpful recommendations, all of which have enabled us to enhance the text's quality.

Reviewer # 1

Comment no: 1 Conclusion parts in both the abstract and the main text needs to be rewritten and lengthened. Consider adding more supporting evidence.

Reply no: 1 Thank you for your comment. We have now upgraded the conclusion section, both in the abstract and in the main text, and we hope it fulfills your expectations.

Comment no: 2 The discussion should be more comprehensive. Furthermore, the introduction presents an appropriate outline but it is better to be lengthened. Adding additional references is recommended in both introduction and discussion section.

Reply no: 2 Thank you for your suggestion. In accordance with your suggestion, we have incorporated a total of seven references into the introduction sections (2, 8, 14, 15) and the discussion sections (30, 31, 32, 33, 36, and 37).

Comment no: 3 It is recommended to add a new section at the end of the article after conclusion section, describing any abbreviation which is being used throughout the article.

Reply no: 3 Thank you for your comment. In the revised text, we added a new section describing the abbreviation after the acknowledgement.

Reviewer # 2

I am glad to have the opportunity to review this manuscript and thank the authors for conducting this research on the use of intermittent pneumatic compression therapy as a preventive measure for venous thromboembolism after total hip arthroplasty. The results section lacks clarity on the added value of the study to the topic. The authors should provide conclusive insights from the included studies and consider presenting new findings. Additionally, the discussion requires substantial revision. My specific comments are as follows:

Comment no: 1 Line 51: The authors should provide a full explanation of "THA" when it is first mentioned in the manuscript.

Reply no: 1 Thank you for your comment. Line 27 in the abstract section already mentions the abbreviation of THA (Total Hip Arthroplasty).

Comment no: 2 Line 53: Consider replacing the word “lakh” with a term more universally understood by the audience.

Reply no: 2 We replaced the word lakhs with million in the revised text.

Comment no: 3 Line 55: The authors are encouraged to explain "VTE" the first time it appears in the manuscript.

Reply no: 3 Added in the revised text.

Comment no: 4 Line 71: Similarly, please provide an explanation for "IPC" at its first mention in the manuscript.

Reply no: 4 Added in the revised text.

Comment no: 5 Line 91: Could the authors clarify the rationale behind restricting the search dates to studies published from 2004 onward?

Reply no: 5 Thank you for your comment. Our intention was to gather evidence from the past two decades regarding the use of devices.

Comment no: 6 Line 94: The complete search strategy should be included as a supplementary file and referenced in the corresponding paragraph of the manuscript. If this is not feasible, please provide it in your response to comments.

Reply no: 6 Thank you for your comment. The revised text now segregates Table 1 as supplementary file 2.

Comment no: 7 Line 105: The authors state that RCTs were excluded; however, the results section mentions that all included articles were RCTs. This discrepancy suggests an error in line 105 that needs correction.

Reply no: 7 We intended to mention that “randomized controlled designs published in languages other than English also excluded” which is now corrected in the revised text.

Comment no: 8 Line 109: In the “Selection Process” section, please specify how differences between the two reviewers were resolved.

Reply no: 8 Thank you for your comments. We resolved differences between two reviewers using predesigned inclusion and exclusion criteria.

Comment no: 9 Line 150, Figure 2: The authors should address the selective bias and reporting bias components in Figure 2 for completeness.

Reply no: 9 We appreciate your inquiry and the chance to enhance the manuscript's quality. As previously stated: “In all papers examined, the author did not acknowledge any kind of reporting bias or other biases.” As a result, we identified an ambiguous risk bias associated with two variables: inadequate data outcomes and reporting bias. That’s why we left it blank.

Comment no: 10 Table 4: It would enhance the table's informativeness if study design, country, and follow-up duration were included.

Reply no: 10 The text already mentions the study design, country information, and population characteristics. Table 4 also mentions the duration of treatment under the heading Intervention. However, the studies do not specify the duration of follow-up, so we are unable to include them.

Comment no: 11 Table 4: Two of the included studies were conducted by Paudel et al. The authors are advised to verify whether these two articles share participants. If they do, only the more comprehensive study should be included unless the outcomes reported are distinct.

Reply no: 11 Thank you for your comment. However, the same author conducted both studies in different geographical areas (India and Nepal), with varying participant numbers. For this reason, we have included both studies in our review.

Comment no: 12 Line 183: "LMWH" should be explained the first time it is mentioned in the manuscript.

Reply no: 12 Mentioned in the revised text.

Comment no: 13 Results Section: The results section lacks clarity on the added value of the study to the topic. The authors should provide conclusive insights from the included studies and consider presenting new findings, such as the VTE rate following IPC, using proportion analysis or pooled prevalence analysis.

Reply no: 13 Our aim was to conduct a systematic review regarding this issue. However, in the future, we could potentially conduct a meta-analysis, such as a proportion analysis or a pooled prevalence analysis, to address this issue.

Comment no: 14 Line 227: The definition of IPC appears in the discussion section but should be introduced earlier, in the introduction, where IPC is first mentioned.

Reply no: 14 Addressed in the revised text.

Comment no: 15 Discussion Section: The discussion requires substantial revision. Much of its content (e.g., Lines 208-214, Lines 216-223, Lines 224-246) would be more appropriate in the results section. The discussion should focus on synthesizing the findings, their implications, and the broader context of the study.

Reply no: 15 I appreciate your feedback. The material of the discussion area has been upgraded to perhaps meet your criteria.

Comment no: 16 General Comment: The manuscript would benefit from a thorough revision of its English for clarity and readability.

Reply no: 16 I appreciate your feedback. We have updated the manuscript and enhanced the general quality of the English for clarity and readability.

---

## [Decision Letter · Decision Letter 1]

24 Jan 2025

Intermittent pneumatic compression therapy as a preventive measure for venous thromboembolism after total hip arthroplasty: A systematic review

PONE-D-24-45814R1

Dear Dr. Ahmed,

We’re pleased to inform you that your manuscript has been judged scientifically suitable for publication and will be formally accepted for publication once it meets all outstanding technical requirements.

Kind regards,

Amirhossein Ghaseminejad-Raeini

Academic Editor

PLOS ONE

Additional Editor Comments (optional):

Reviewers' comments:

Reviewer's Responses to Questions

**Comments to the Author**

1. If the authors have adequately addressed your comments raised in a previous round of review and you feel that this manuscript is now acceptable for publication, you may indicate that here to bypass the “Comments to the Author” section, enter your conflict of interest statement in the “Confidential to Editor” section, and submit your "Accept" recommendation.

Reviewer #1: All comments have been addressed

Reviewer #2: (No Response)

2. Is the manuscript technically sound, and do the data support the conclusions?

Reviewer #1: Yes

Reviewer #2: (No Response)

3. Has the statistical analysis been performed appropriately and rigorously? 

Reviewer #1: Yes

Reviewer #2: (No Response)

4. Have the authors made all data underlying the findings in their manuscript fully available?

Reviewer #1: Yes

Reviewer #2: (No Response)

5. Is the manuscript presented in an intelligible fashion and written in standard English?

Reviewer #1: Yes

Reviewer #2: (No Response)

6. Review Comments to the Author

Reviewer #1: The manuscript is well-structured and well-analyzed and provides valuable results. The tables

and figures are appropriately depicted and Cleary presented. Study limitation are mentioned in a

separate section in detail, adding additional strength to the article. The Discussion section

required to be more thorough and detailed and in the revised version it has been expanded and

more references are incorporated.

At the end I have some questions about this article.

The specific questions and recommendations are as follows:

1:

Conclusion parts in both the abstract and the main text needs to be rewritten and lengthened. Consider

adding more supporting evidence.

Conclusion part in the revised version is more informative than before, and clinical implications are also

better discussed.

2:

The discussion should be more comprehensive. Furthermore, The introduction presents an appropriate

outline but it is better to be lengthened. Adding additional references is recommended in both introduction

and discussion section.

Appropriately revised by authors, an additional 7 new references has been incorporated in the new

version.

3:

It is recommended to add a new section at the end of the article after conclusion section, describing any

abbreviation which is being used throughout the article.

A new table describing the any abbreviated term has been added properly.

Thanks to the authors for their hard work on this Revision. It’s a pleasure to review such a contribution to

the field.

Reviewer #2: (No Response)

7. PLOS authors have the option to publish the peer review history of their article (what does this mean? ). If published, this will include your full peer review and any attached files.

**Do you want your identity to be public for this peer review?** For information about this choice, including consent withdrawal, please see our Privacy Policy .

Reviewer #1: **Yes: ** SEYED AMIRHOSSEIN MAZHARI

Reviewer #2: No

---

## [Editor Report · Acceptance letter]

PONE-D-24-45814R1

PLOS ONE

Dear Dr. Ahmed,

I'm pleased to inform you that your manuscript has been deemed suitable for publication in PLOS ONE. Congratulations! Your manuscript is now being handed over to our production team.

Kind regards,

on behalf of

Dr. Amirhossein Ghaseminejad-Raeini

Academic Editor

PLOS ONE